# On Predictive Information Sub-optimality of RNNs

## Abstract

Certain biological neurons demonstrate a remarkable capability to optimally compress the history of sensory inputs while being maximally informative about the future. In this work, we investigate if the same can be said of artificial neurons in recurrent neural networks (RNNs) trained with maximum likelihood. In experiments on two datasets, restorative Brownian motion and a hand-drawn sketch dataset, we find that RNNs are sub-optimal in the information plane. Instead of optimally compressing past information, they extract additional information that is not relevant for predicting the future. We show how constraining past information by injecting noise into the hidden state can improve the ability of RNNs to extract predictive information for both maximum likelihood and contrastive loss training.

## 1 Introduction

Remembering past events is a critical component of predicting the future and acting in the world. An information-theoretic quantification of how much observing the past can help in predicting the future is given by the *predictive information* (Bialek et al., 2001). The predictive information is the mutual information (MI) between a finite set of observations (the *past* of a sequence) and an infinite number of additional draws from the same process (the *future* of a sequence). As a mutual information, the predictive information gives us a reparameterization independent, symmetric, interpretable measure of the co-dependence of two random variables. More colloquially, the mutual information tells us how many *bits* we can predict of the future given our observations of the past.[1] Asymptotically, a vanishing fraction of the information in the past is relevant to the future (Bialek et al., 2001), thus systems which excel at prediction need not memorize the entire past of a sequence.

Intriguingly, certain biological neurons extract representations that efficiently capture the predictive information in sequential stimuli (Palmer et al., 2015; Tkačik & Bialek, 2016). In Palmer et al. (2015), spiking responses of neurons in salamander retina had near optimal mutual information with the *future* states of sequential stimuli they were exposed to, while compressing the *past* as much as possible.

Do our artificial neural networks perform similarly? In this work, we aim to answer this question by measuring artificial recurrent neural networks' (RNNs) capacity to compress the past while retaining relevant information about the future.

Our contributions are as follows:

- We demonstrate that RNNs are suboptimal at extracting predictive information on the tractable sequential stimuli used in Palmer et al. (2015).
- We thoroughly validate the accuracy of our mutual information estimates on RNNs and optimal models.
- We repeat our analysis on the real-world Aaron Koblin Sheep sketch dataset (Ha & Eck, 2017), demonstrating that deterministic RNNs continue to perform suboptimally.
- We demonstrate that on small datasets, models trained to compress the past have improved sample quality over naively-trained RNNs.

---

[1] Because of its symmetry, this is equivalent to the number of bits we can reconstruct of the past given observations of the future.

## 2 BACKGROUND AND METHODS

We begin by providing additional background on predictive information, mutual information estimators, stochastic RNNs, and Gaussian Information Bottleneck. These tools are neccessary for accurately evaluating the question of whether RNNs are optimal in the information plane, as we require knowledge of the optimal frontier, and accurate estimates of mutual information for complex RNN models.

### 2.1 PREDICTIVE INFORMATION

Imagine an infinite sequence of data: $X_t$. The predictive information (Bialek et al., 2001) in the sequence is the mutual information between some finite number of observations of the past ($T$) and the infinite future of the sequence:

$$I_{\text{pred}}(T) = I(X_{\text{past}}; X_{\text{future}}) = I(\{X_{t-T+1}, \ldots, X_t\}; \{X_{t+1}, \ldots\}). \tag{1}$$

For a process for which the dynamics are not varying in time, this will be independent of the particular time $t$ chosen to be the present. More specifically, the predictive information is an expected log ratio between the likelihood of observing a future given the past, versus observing that future in expectation over all possible pasts:

$$I_{\text{pred}} \equiv \mathbb{E}\left[\log \frac{p(x_{\text{future}}|x_{\text{past}})}{p(x_{\text{future}})}\right] = \mathbb{E}\left[\log \frac{p(x_{\text{future}}|x_{\text{past}})}{\mathbb{E}_{x'_{\text{past}}}\left[p(x_{\text{future}}|x'_{\text{past}})\right]}\right]. \tag{2}$$

A sequential model such as an RNN provides a stochastic representation of the entire past of the sequence $Z \sim p(z|x_{\text{past}})$. For any such representation, we can measure how much information it retains about the past, a.k.a. the *past information*: $I_{\text{past}} = I(Z; X_{\text{past}})$, and how informative it is about the future, a.k.a. the *future information*: $I_{\text{future}} = I(Z; X_{\text{future}})$. Because the representation depends only on the past, our three random variables satisfy the Markov relations: $Z \leftarrow X_{\text{past}} \leftrightarrow X_{\text{future}}$ and the Data Processing Inequality (Cover & Thomas, 2012) ensures that the information we have about the future is always less than or equal to both the true predictive information of the sequence ($I_{\text{future}} \leq I_{\text{pred}}$) as well as the information we retain about the past ($I_{\text{future}} \leq I_{\text{past}}$). For any particular sequence, there will be a frontier of solutions that optimally tradeoff between $I_{\text{past}}$ and $I_{\text{future}}$. A common method for tracing out this frontier is through the Information Bottleneck (Tishby et al., 2000) Lagrangian:

$$\min_{p(z|x_{\text{past}})} I(Z; X_{\text{past}}) - \beta I(Z; X_{\text{future}}), \tag{3}$$

where the parameter $\beta$ controls the tradeoff. An *efficient* representation of the past is one that lies on this *optimal frontier*, or equivalently is a solution to Eqn. 3 for a particular choice of $\beta$. For simple problems, where the sequence is jointly Gaussian, we will see that the optimal frontier can be identified analytically.

### 2.2 MUTUAL INFORMATION ESTIMATORS

In order to measure whether a representation is efficient, we need a way to measure its past and future informations. While mutual information estimation is difficult in general (McAllester & Stratos, 2019), recent progress has been made on a wide range of variational bounds on mutual information (Poole et al., 2019). While these provide bounds and not exact estimates of mutual information, they allow us to compare mutual information quantities in continuous spaces across models. There are two broad families of estimators: variational lower bounds powered by a tractable generative model, or contrastive lower bounds powered by an unnormalized critic.

The former class of lower bounds, first presented in Barber & Agakov (2003), are powered by a variational generative model:

$$I_{\text{future}} = \mathbb{E}\left[\log \frac{p(x_{\text{future}}|z)}{p(x_{\text{future}})}\right] \geq H(x_{\text{future}}) + \mathbb{E}\left[\log q(x_{\text{future}}|z)\right]. \tag{4}$$

A generative model provides a demonstration that there exists at least some information between the representation $z$ and the future of the sequence. For our purposes, $H(x_{\text{future}})$, the entropy of the

future of the sequence is a constant, determined by the dynamics of the sequence itself and outside our control. For tractable problems, such as the toy problem we investigate below, this value is known. For real datasets, this value is not known so that we cannot produce reliable estimates of the mutual information. It does, however, still provide reliable gradients of a lower bound. One example of such a generative model is the loss used to train the RNN to begin with.

Contrastive lower bounds can be used to estimate $I_{\text{future}}$ for datasets where building a tractable generative model of the future is challenging. InfoNCE style lower bounds (van den Oord et al., 2018; Poole et al., 2019) only require access to *samples* from both the joint distribution and the individual marginals:

$$I(X; Z) \geq I_{\text{NCE}}(X; Z) \triangleq \mathbb{E}_{p^K(x,z)} \left[ \frac{1}{K} \sum_{i=1}^{K} \log \frac{e^{f(x_i, z_i)}}{\frac{1}{K} \sum_{j=1}^{K} e^{f(x_j, z_i)}} \right]. \tag{5}$$

Here $f(x_j, z_i)$ is a trained *critic* that plays a role similar to a discriminator in a Generative Adversarial Network (Goodfellow et al., 2014). It scores pairs, attempting to determine if an $(x, z)$ pair came from the joint ($p(x, z)$) or the factorized marginal distributions ($p(x)p(z)$).

When forming estimates of $I_{\text{past}}$, we can leverage additional knowledge about the known encoding distribution from the stochastic RNN $p(z|x_{\text{past}})$ to form tractable upper and lower bounds without having to learn an additional critic (Poole et al., 2019):

$$\mathbb{E}\left[ \frac{1}{K} \sum_{i=1}^{K} \log \frac{p(z^i|x_{\text{past}}^i)}{\frac{1}{K} \sum_j p(z^i|x_{\text{past}}^j)} \right] \leq I(Z; X_{\text{past}}) \leq \mathbb{E}\left[ \frac{1}{K} \sum_{i=1}^{K} \log \frac{p(z^i|x_{\text{past}}^i)}{\frac{1}{K-1} \sum_{j \neq i} p(z^i|x_{\text{past}}^j)} \right]. \tag{6}$$

We refer to these bounds as *minibatch upper and lower bounds* as they are computed using minibatches of size $K$ from the dataset. As the minibatch size $K$ increases, the upper and lower bounds can become tight. When $\log K \ll I(Z; X_{\text{past}})$ the lower bound saturates at $\log K$ and the upper bound can be loose, thus we require using large batch sizes to form accurate estimates of $I_{\text{past}}$.

## 2.3 STOCHASTIC RNNS

Deterministic RNNs can theoretically encode infinite information about the past in their hidden states (up to floating point precision). To limit past information, we devise a simple stochastic RNN. Given the deterministic hidden state $h_t$, we output a stochastic variable $z_t$ by adding i.i.d. Gaussian noise to the hidden state before reading out the outputs: $z_t \sim \mathcal{N}(h_t, \sigma^2)$. These stochastic outputs are then used to predict the future state: $\hat{x}_{t+1} \sim \mathcal{N}(g_{\text{decoder}}(z_t), \sigma_o^2)$, as illustrated in Figure 1. With bounded activation functions on the hidden state $h_t$, we can use $\sigma^2$ to upper bound the information stored about the past in the stochastic latent $z_t$. This choice of stochastic recurrent model yields a tractable conditional distribution $p(z_t|x_{\leq t}) \sim \mathcal{N}(h_t, \sigma^2)$, which we can use in Eq. 6 to form tractable upper and lower bounds on the past information. We will consider two different settings for our stochastic RNNs: (1) where the RNNs are trained deterministically and the noise on the hidden state is added only at evaluation time, and (2) where the RNNs are trained with noise, and evaluated with noise.

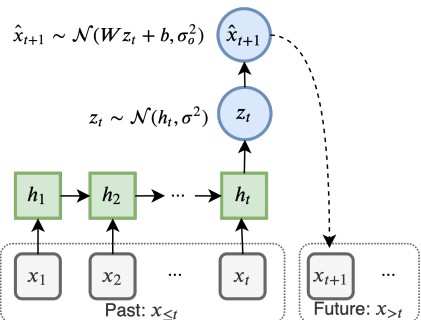

Figure 1: Schematic of Gaussian-noise-augmented stochastic RNN.

## 2.4 GAUSSIAN INFORMATION BOTTLENECK

To evaluate optimality of RNNs, we first focus on the tractable sequential stimuli of a simple Brownian Harmonic Oscillator as used in (Palmer et al., 2015). A crucial property of this dataset is that we can analytically calculate the optimal trade-off between past and future information as it is an instance of the Gaussian Information Bottleneck (Chechik et al., 2005).

Consider jointly multivariate Gaussian random variables $X \in \mathbb{R}^{D_X}$ and $Y \in \mathbb{R}^{D_Y}$, with covariance $\Sigma_X$ and $\Sigma_Y$ and cross-covariance $\Sigma_{XY}$. The solution to the Information Bottleneck objective:

$$\min_T I(X;T) - \beta I(Y;T),\tag{7}$$

is given by a linear transformation $T = \boldsymbol{A}X + \varepsilon$ with $\varepsilon \sim \mathcal{N}(\boldsymbol{0}, \Sigma_\varepsilon)$. The projection matrix $A$ projects along the lowest eigenvectors of $\Sigma_{X|Y}\Sigma_X^{-1}$, where the trade-off parameter $\beta$ decides how many of the eigenvectors participate. Further details can be found in Appendix A.1.

## 3 RESULTS

### 3.1 BROWNIAN HARMONIC OSCILLATOR

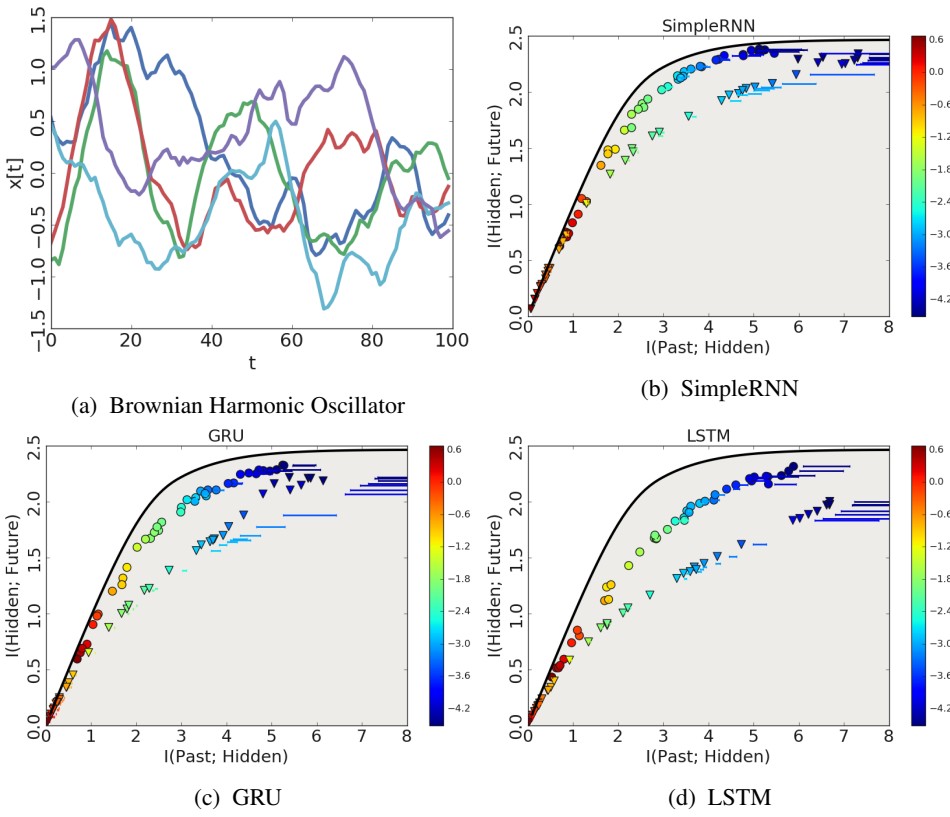

(a) Brownian Harmonic Oscillator

(b) SimpleRNN

(c) GRU

(d) LSTM

Figure 2: (*a*) Example trajectories of Brownian harmonic oscillator over time, with each color representing a different trajectory. (*b, c, d*) Estimates of the past information contained in the stochastic variable (x-axis) vs. future information (y-axis). The feasible region is shaded. The models to the upper-left perform better future prediction, while compressing the past more, therefore they are of higher efficiency. Points correspond to estimates with a learned critic; bars represent the gap between the lower and upper bounds using the known conditional distribution instead of a learned critic; colors represent the noise level, $\log(\text{noise})$, added to the hidden state. The stochastically trained RNN is marked as ∘, and deterministically trained RNN with post-hoc noise injection is marked as ▽.

We begin by considering the Brownian harmonic oscillator (BHO) dataset, a form of restorative Brownian motion, as in Palmer et al. (2015). This system has the benefit of having fully tractable

predictive information. The dynamics are given by:

$$
\begin{aligned}
x_{t+\Delta t} &= x_t + v_t \Delta t, \\
v_{t+\Delta t} &= [1 - \Gamma \Delta t] v_t - \omega^2 x_t \Delta t + \xi_t \sqrt{D \Delta t}.
\end{aligned}
\tag{8}
$$

where $\xi_t$ is a standard Gaussian random variable. Additional details can be found in Appendix A.2. Examples of the trajectories for this system can be seen in Figure 2a.

Given its analytical tractability we can explicitly compare the estimated RNN performance against optimal performance. We compared three major variants of RNNs, including fully-connected RNNs, gated recurrent units (GRU, Cho et al. (2014)), and LSTMs (Hochreiter & Schmidhuber, 1997). Each network had 32 hidden units and `tanh` activations. Full training details are in Appendix A.2.

Figure 2 contains the results for both stochastic RNNs and deterministic RNNs with noise added only at evaluation time. By varying the strength of the noise, networks trace out a trajectory on the information plane. We find that networks trained with noise are close to the optimal frontier (black), in terms of optimally extracting information about the past that is useful for predicting the future. While injecting noise at evaluation time produces networks with compressed representations, these deterministically trained networks perform worse than their stochastically trained counterparts.

Training with noise injection can make the networks more efficiently capture the predictive information than post-hoc noise injection. At the same noise level (the color coding in Figure 2) stochastically trained RNNs have both higher $I(z; x_{\text{past}})$ and higher $I(z; x_{\text{future}})$. By limiting the capacity of the models during training, they were able to extract more efficient representations. For this task, we find that more complex RNN variants such as LSTMs are less efficient at encoding predictive information.

Dropout (Srivastava et al., 2014) is a potential alternative way to eliminate information. In Appendix A.5, we demonstrate that RNNs trained with dropout extract less information than the ones trained without dropout, and that our simple noise injection technique can find equivalent models. To access whether the observed sub-optimality was a consequence of our choice of maximum likelihood (MLE) as a training objective, in Appendix A.4 we demonstrate the effect of using contrastive predictive coding (CPC) (Oord et al., 2018) as the training objective. We find that models trained with CPC loss perform similarly to models trained with MLE, as measured on the information plane. However, this may be due to the BHO dataset having Markovian dynamics, and optimizing for one-step-ahead prediction with MLE is sufficient to maximize mutual information with the future of the sequence.

## 3.2 Accuracy of Mutual Information Estimates

Our claim that RNNs are suboptimal in capturing predictive information hinges on the quality of our MI estimates. Are the RNNs truly suboptimal or is it just that our MI estimates are inaccurate? Here we provide several experiments further validating the accuracy of our MI estimates.

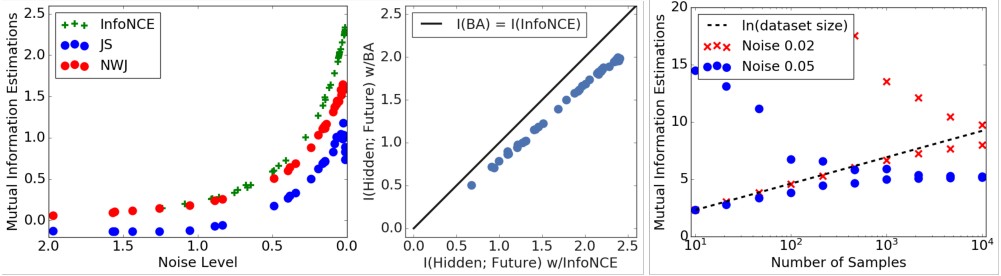

Figure 3: **(Left)** Comparison among critic based estimators, InfoNCE, JS and NWJ. **(Middle)** Comparison between estimations from Barber-Agakov and InfoNCE lower bound on future information $I(X_{\text{future}}; Z)$. **(Right)** Illustration of the convergence for minibatch upper and lower bounds with two noise levels, $0.02$ (Blue) and $0.05$ (Red). Dashed line is the $\ln$(number of samples), which is the limit for minibatch lower bounds.

**Comparison among estimators.** There are many different mutual information estimators. In Figure 3, we compare various mutual information lower bounds with learned critics: InfoNCE, NWJ and JS, as summarized in Poole et al. (2019). NWJ and JS show higher variance and worse bias than InfoNCE. The second panel of Figure 3 demonstrates that InfoNCE outperforms a variational Barber-Agakov style variational lower bound at measuring the future information. Therefore, we adopted InfoNCE as the critic based estimator for the future information in the previous section.

For the past information, we could generate both tractable upper and lower bounds, given our tractable likelihood, $p(z_t|x_{\leq t}) \sim \mathcal{N}(h_t, \sigma^2)$. In the third panel of Figure 3 we demonstrate that these bounds become tight as the sample size increases. However they require a large number of samples before they converge. Fundamentally, the lower bound itself is upper-bounded by the log of the number of samples used.

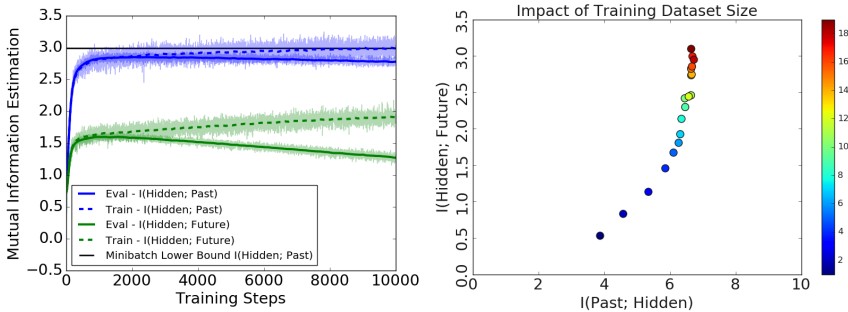

Figure 4: **(Left)** Estimates of past and future information over training iterations on the training and testing BHO data. We can see that our MI estimates quickly overfit, which we remedy here by early stopping. **(Right)** Impact of training dataset sizes on InfoNCE estimator on more complicated dataset, Aaron's Sheep, as introduced in Section 3.3. The original dataset has size 7200 for training, and 800 for evaluation. We augment the dataset by random scaling the input values per sequence. The colors indicate the multiples of original dataset size after augmentation.

**Estimator training with finite dataset.** Training the learned critic on finite datasets for a large number of iterations resulted in problematic memorization and overestimates of MI. To counteract the overfitting, we performed early stopping using the MI estimate with the learned critic on a validation set. Unlike the training MI, this is a valid lower bound on the true MI. We then report estimates of mutual information using the learned critic on an independent test set, as Figure 4.

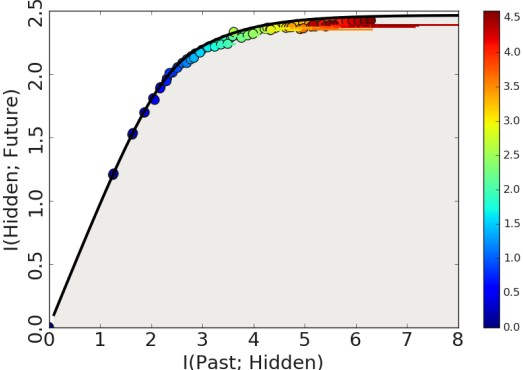

Figure 5: Evaluate mutual information estimators given optimal encoder. For past information estimation, I(Hidden; Past), InfoNCE lower bound (colored points) and MB-Lower and MB-Upper (colored bars) are used; for future information, I(Hidden; Future), only InfoNCE is applied, due to lack of tractable conditional distribution $p(Y|T)$. Heat-color represents the level of trade-off parameter $\beta$.

**Analytical Agreement** As a final and telling justification of the efficiency of our estimators, Figure 5 demonstrates that our estimators accurately estimate the mutual information in our region of interest,

for analytically derived optimal projections. Background for the Gaussian Information Bottleneck is in Section 2.4 and details of the calculation can be found in Appendix A.1.

### 3.3 Vector Drawing Dataset

In order to assess whether the sub-optimality of RNNs was an artifact of the BHO dataset, we performed additional experimentson a real world sequential dataset. We used the Aaron Koblin Sheep Sketch Dataset[2]. Full experimental details are in Appendix A.3. We adopted the SketchRNN architecture and online data augmentation procedure used in Ha & Eck (2017).

Figure 6 (Left) shows the estimates on the information plane for the trained networks. Again, the networks that were trained explicitly with noise dominate on the information plane. For this real world task we no longer know the optimal information tradeoff, but still see that the deterministically trained networks evaluated with noise are demonstrably suboptimal compared to some simple stochastic networks trained with noise injection.

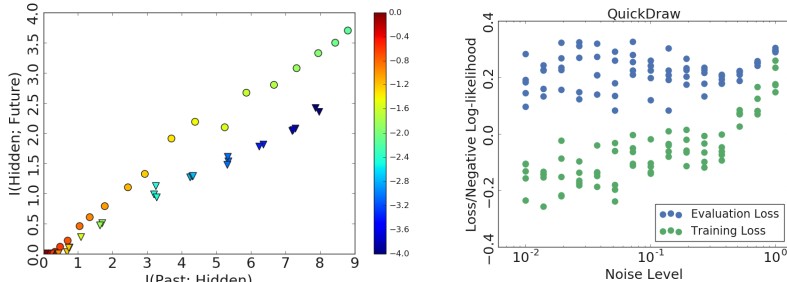

Figure 6: **(Left)** Evaluation on Aaron Sheep dataset by comparing training explicitly with noise ($\circ$) and post-hoc noise injection after training ($\triangledown$). The color bar shows the noise level in $\log_{10}$ scale. **(Right)** Comparing the training and evaluation loss for noise-trained RNNs.

Besides demonstrating sub-optimality in capturing predictive information on the information plane, in Figure 6 (Right), we demonstrate that the deterministic RNNs are wasteful. Networks with higher levels of compression (marked by higher noise levels) were able to obtain similar average test set performance with noticeably lower variance than the deterministic networks. They also show a smaller generalization gap. By explicitly training the networks to operate within some constrained information budget reduced variance between runs without substantially reducing test set performance.

In Figure 7 we show some generated completions from the models trained on the full dataset with varying degrees of noise. The models trained with the highest noise levels show a noticeable degradation in sample quality.

### 3.4 Small Datasets

We expect the benefits of training compressed representations to be most marked on limited datasets where the compression can prevent the network from memorizing too many spurious correlations.

To investigate, we repeated the experiments of the previous section with limited dataset with only 100 examples. Figure 8 (Upper Left) shows the corresponding information plane points for stochastically trained RNNs with various noise levels. Notice that at about 4 nats of past information, the networks future information essentially saturates. While it seems as though the networks don't suffer even when asked to learn richer representations, this is largely an artifact of our training procedure which included early stopping. As can be seen in Figure 8 (Lower Left), all of our networks overfit in terms of evaluation loss, but the onset of overfitting was strongly controlled by the degree of compression.

---

[2]Available from `https://github.com/hardmaru/sketch-rnn-datasets/tree/master/aaron_sheep`

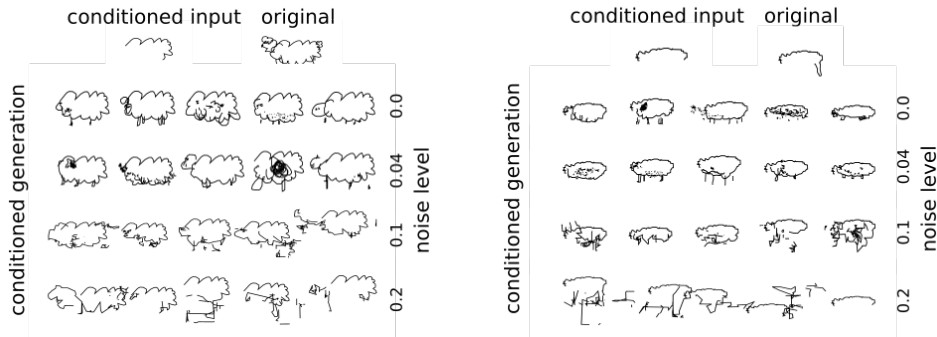

Figure 7: Conditional generation of QuickDraw sketches by SketchRNN models trained with different noise levels.

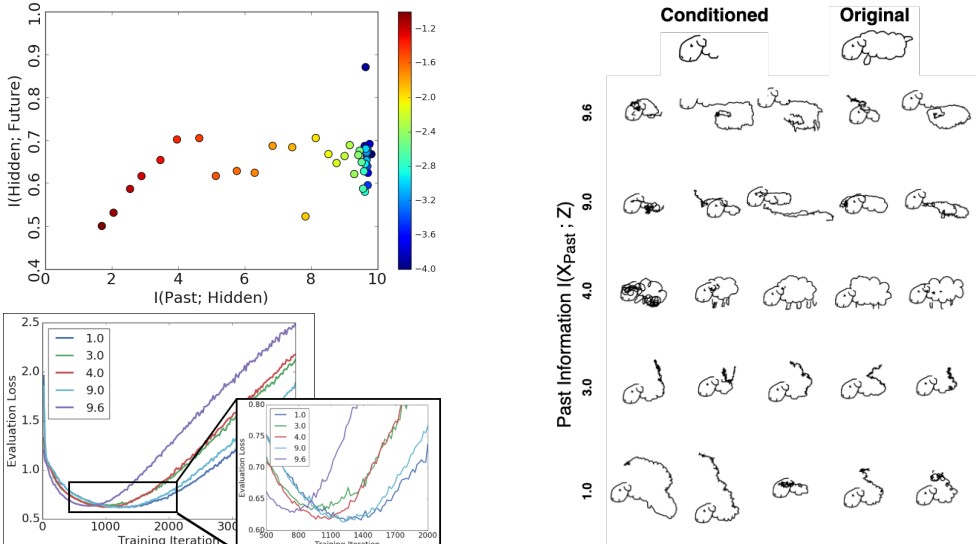

Figure 8: **Left Column.** Top: Estimation of past and future information for RNN trained with $100$ samples, with color indicating the noise levels in $\log_{10}$ scale. Bottom: Validation loss for different noise levels. **Right Column.** Conditional generated samples from models with different levels of past information. The generation is conditioned on a 25-step stroke, which is taken from a held-out sample. The samples from the model with $4.0$ nats past information is of better sample quality than models with higher past informations. For more samples, see Figure 12.

Most noticeably, in the limited data regime, compressed representations lead to improved sample quality, as seen in Figure 8 (Right). Models with intermediately-sized compressed representations show the best generated samples while retaining a good amount of diversity. Models with either too little or too much past information tend to produce nonsensical generations.

## 4 CONCLUSION AND DISCUSSION

In this work, we have demonstrated how analyzing recurrent neural networks in terms of predictive information can be a useful tool in probing and understanding behavior. We find that RNNs trained with maximum likelihood are sub-optimal on the information plane, extracting more information about the past than is required to predict the future. By analyzing different training objectives and noise injection approaches in the information plane, we can better understood the tradeoffs made by different models, and identify models that are closer to the optimality demonstrated by biological neurons (Palmer et al., 2015).

While the simple strategy of adding noise to a bounded hidden state can be used to constrain information, setting the amount of noise and identifying where one should be on the information plane remains an open problem. Additionally, studying the impact of other architecture choices, such as stochastic latent variables in variational RNNs (Chung et al., 2015), or attention-based Transformers (Vaswani et al., 2017) in the information plane could yield insights into their better performance on several tasks.

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

# A   APPENDIX

## A.1   GAUSSIAN INFORMATION BOTTLENECKS

Consider jointly multivariate Gaussian random variables $X \in \mathbb{R}^{D_X}$ and $Y \in \mathbb{R}^{D_Y}$, with covariance $\Sigma_X$ and $\Sigma_Y$ and cross-covariance $\Sigma_{XY}$. The solution to the Information Bottleneck objective:

$$\min_{T} I(X;T) - \beta I(Y;T), \tag{9}$$

is given by a linear transformation $T = \boldsymbol{A}X + \varepsilon$ with $\varepsilon \sim \mathcal{N}(\boldsymbol{0}, \Sigma_\varepsilon)$. The projection matrix $A$ projects along the lowest eigenvectors $\lambda_i (i \in [1, D_X])$ of $\Sigma_{X|Y}\Sigma_X^{-1}$, where the trade-off parameter $\beta$ decides how many of the eigenvectors participate, $\mathbf{v}_i^T (i \in [1, D_X])$. The projection matrix $\boldsymbol{A}$ could be analytically derived as

$$\boldsymbol{A} = [\alpha_{\mathbf{1}}\mathbf{v}_{\mathbf{1}}^{\mathbf{T}}, \alpha_{\mathbf{2}}\mathbf{v}_{\mathbf{2}}^{\mathbf{T}}, \dots, \alpha_{\mathbf{D_X}}\mathbf{v}_{\mathbf{D_X}}^{\mathbf{T}}], \ \Sigma_\varepsilon = \mathbb{I} \tag{10}$$

where the projection coefficients $\alpha_{\mathbf{i}}^{\mathbf{2}} = \max(\frac{\beta(1-\lambda_i)-1}{\lambda_i r_i}, 0)$, $r_i = \mathbf{v}_{\mathbf{i}}^{\mathbf{T}}\boldsymbol{\Sigma}_{\mathbf{X}}\mathbf{v}_{\mathbf{i}}$, with proof in Appendix A.1.1.

Given the optimally projected states $T$, the *optimal frontier* (black curve in Figure 5) is:

$$I(T;Y) = I(T;X) - \frac{n_I}{2}\log\left(\prod_{i=1}^{n_I}(1-\lambda_i)^{\frac{1}{n_I}} + e^{\frac{2I(T;X)}{n_I}}\prod_{i=1}^{n_I}\lambda_i^{\frac{1}{n_I}}\right), \ c_{n_I} \le I(T;X) \le c_{n_I+1} \tag{11}$$

where $n_I$ is the cutoff number indicating the number of smallest eigenvalues being used. The critical points $c_{n_I}$, changing from using $n_I = N$ eigenvalues to $N + 1$ eigenvalues, can be derived given the concave and $C^1$ smoothness property for the optimal frontier, with proof in Appendix A.1.2:

$$c_{n_I} = \frac{1}{2}\sum_{i=1}^{N}\log\frac{\lambda_{N+1}}{\lambda_i}\frac{1-\lambda_i}{1-\lambda_{N+1}} \tag{12}$$

### A.1.1 PROOF OF OPTIMAL PROJECTION

By Theorem 3.1 of Chechik et al. (2005), the projection matrix for optimal projection is given by

$$
\boldsymbol{A} = \left\{
\begin{array}{ll}
\left[ \alpha_1 \mathbf{v_1^T}, \mathbf{0}, \mathbf{0}, \ldots, \mathbf{0} \right], & 0 \le \beta \le \beta_1 \\
\left[ \alpha_1 \mathbf{v_1^T}, \alpha_2 \mathbf{v_2^T}, \mathbf{0}, \ldots, \mathbf{0} \right], & \beta_1 \le \beta \le \beta_2 \\
\left[ \alpha_1 \mathbf{v_1^T}, \alpha_2 \mathbf{v_2^T}, \alpha_2 \mathbf{v_2^T}, \ldots, \mathbf{0} \right], & \beta_2 \le \beta \le \beta_3 \\
\vdots
\end{array}
\right\}
\tag{13}
$$

where $\mathbf{v}_i^T (i \in [1, D_X])$ are left eigenvectors of $\Sigma_{X|Y} \Sigma_X^{-1}$ sorted in ascending order by the eigenvalues $\lambda_i (i \in [1, D_X])$; $\beta_i = \frac{1}{1-\lambda_i}$ are critical values for trade-off parameter $\beta$; and the projection coefficients are $\alpha_{\mathbf{i}}^{\mathbf{2}} = \frac{\beta(1-\lambda_i)-1}{\lambda_i r_i}$, $r_i = \mathbf{v_i^T} \Sigma_X \mathbf{v_i}$. In practice, noticing that $\beta * (1-\lambda_i) - 1 < 0$ when $\beta < \beta_i$, we simplify Equation (13) as $\boldsymbol{A} = [\alpha_1 \mathbf{v_1^T}, \alpha_2 \mathbf{v_2^T}, \ldots, \alpha_{\mathbf{D_X}} \mathbf{v_{D_X}^T}]$ with $\alpha_{\mathbf{i}}^{\mathbf{2}} = \max(\frac{\beta(1-\lambda_i)-1}{\lambda_i r_i}, 0)$.

### A.1.2 PROOF OF CRITICAL POINTS ON OPTIMAL FRONTIER

By Eq.15 of Chechik et al. (2005)

$$
I(T;Y) = I(T;X) - \frac{n_I}{2} \log \left( \prod_{i=1}^{n_I} (1-\lambda_i)^{\frac{1}{n_I}} + e^{\frac{2I(T;X)}{n_I}} \prod_{i=1}^{n_I} \lambda_i^{\frac{1}{n_I}} \right)
\tag{14}
$$

where $n_I$ is the cutoff on the number of eigenvalues used to compute the bound segment, with eigenvalues sorted in ascending order.

In order to calculate the changing point, where one switching from choosing $n_I = N$ to $N+1$, by $C^1$ smoothness conditions:

$$
\frac{\mathrm{d} I_{n_I=N}(T;Y)}{\mathrm{d} I(T;X)} = \frac{\mathrm{d} I_{n_I=N+1}(T;Y)}{\mathrm{d} I(T;X)}
\tag{15}
$$

LHS is

$$
1 - \frac{\mathrm{d} I_{n_I=N}(T;Y)}{\mathrm{d} I(T;X)} = \frac{\prod_{i=1}^{N} (\lambda_i)^{\frac{1}{N}} e^{\frac{2I(T;X)}{N}}}{\prod_{i=1}^{N} (1-\lambda_i)^{\frac{1}{N}} + e^{\frac{2I(T;X)}{n_I}} \prod_{i=1}^{N} \lambda_i^{\frac{1}{N}}}
\tag{16}
$$

$$
= \frac{e^{\frac{2I(T;X)}{N}}}{e^{\frac{2I(T;X)}{N}} + \prod_{i=1}^{N} \left( \frac{1-\lambda_i}{\lambda_i} \right)^{\frac{1}{N}}}
\tag{17}
$$

Thus, Equation (15) could be rewritten as

$$
\frac{e^{\frac{2I(T;X)}{N}}}{e^{\frac{2I(T;X)}{N}} + \prod_{i=1}^{N} \left( \frac{1-\lambda_i}{\lambda_i} \right)^{\frac{1}{N}}} = \frac{e^{\frac{2I(T;X)}{N+1}}}{e^{\frac{2I(T;X)}{N+1}} + \prod_{i=1}^{N+1} \left( \frac{1-\lambda_i}{\lambda_i} \right)^{\frac{1}{N+1}}}
\tag{18}
$$

Rewrite RHS of above equation, and noticing $\frac{1}{n(n+1)} = \frac{1}{n} - \frac{1}{n+1}$

$$
\frac{e^{\frac{2I(T;X)}{N+1}}}{e^{\frac{2I(T;X)}{N+1}} + \prod_{i=1}^{N+1} \left( \frac{1-\lambda_i}{\lambda_i} \right)^{\frac{1}{N+1}}} = \frac{e^{\frac{2I(T;X)}{N}}}{e^{\frac{2I(T;X)}{N}} + e^{\frac{2I(T;X)}{N(N+1)}} \prod_{i=1}^{N+1} \left( \frac{1-\lambda_i}{\lambda_i} \right)^{\frac{1}{N+1}}}
\tag{19}
$$

The term in lower right corner could be written as

$$\prod_{i=1}^{N+1}(\frac{1-\lambda_i}{\lambda_i})^{\frac{1}{N+1}} \quad = \quad \prod_{i=1}^{N}(\frac{1-\lambda_i}{\lambda_i})^{\frac{1}{N+1}}(\frac{1-\lambda_{N+1}}{\lambda_{N+1}})^{\frac{1}{N+1}} \tag{20}$$

$$= \quad \left(\prod_{i=1}^{N}(\frac{1-\lambda_i}{\lambda_i})^{\frac{1}{N}}\prod_{i=1}^{N}(\frac{\lambda_i}{1-\lambda_i})^{\frac{1}{N(N+1)}}\right)(\frac{1-\lambda_{N+1}}{\lambda_{N+1}})^{\frac{N}{(N+1)N}} \tag{21}$$

$$= \quad \left(\prod_{i=1}^{N}(\frac{1-\lambda_i}{\lambda_i})^{\frac{1}{N}}\right)\left(\prod_{i=1}^{N}\left(\frac{\lambda_i(1-\lambda_{N+1})}{(1-\lambda_i)\lambda_{N+1}}\right)^{\frac{1}{N(N+1)}}\right) \tag{22}$$

Putting RHS together

$$\frac{e^{\frac{2I(T;X)}{N}}}{e^{\frac{2I(T;X)}{N}}+\prod_{i=1}^{N}(\frac{1-\lambda_i}{\lambda_i})^{\frac{1}{N}}} = \frac{e^{\frac{2I(T;X)}{N}}}{e^{\frac{2I(T;X)}{N}}+e^{\frac{2I(T;X)}{N(N+1)}}\left(\prod_{i=1}^{N}(\frac{1-\lambda_i}{\lambda_i})^{\frac{1}{N}}\right)\left(\prod_{i=1}^{N}\left(\frac{\lambda_i(1-\lambda_{N+1})}{(1-\lambda_i)\lambda_{N+1}}\right)^{\frac{1}{N(N+1)}}\right)} \tag{23}$$

To let LHS = RHS, one trivial solution is

$$1 = e^{\frac{2I(T;X)}{N(N+1)}}\left(\prod_{i=1}^{N}\left(\frac{\lambda_i(1-\lambda_{N+1})}{(1-\lambda_i)\lambda_{N+1}}\right)^{\frac{1}{N(N+1)}}\right) \tag{24}$$

Taking $\log$ and cancelling out multiplicative factors, one get the critic point to change from $n_i = N$ to $n_i = N+1$ happens at

$$I(T;X) = \frac{1}{2}\sum_{i=1}^{N}\log\frac{\lambda_{N+1}}{\lambda_i}\frac{1-\lambda_i}{1-\lambda_{N+1}} \tag{25}$$

The original result written in Chechik et al. (2005) missing a factor of $\frac{1}{2}$.

### A.1.3 OPTIMAL PROJECTION

The optimal frontier is generated by joining segments described by Equation (11), as illustrated in Figure 9.

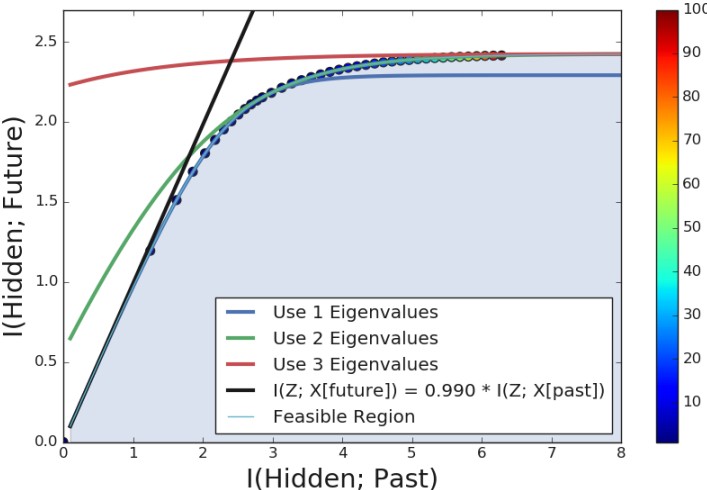

Figure 9: Conditionally generated samples from models with different levels of past information.

## A.2 DETAILS FOR BROWNIAN HARMONIC OSCILLATOR

To generate the sample trajectories, we set the undamped angular velocity $\omega = 1.5 \times 2\pi (\text{rad})$, damping coefficient $\Gamma = 20.0$, and the dynamical range of external forces $D = 1000.0$, with integration time step-size $\Delta t = 0.01667$. The stationary distribution of Equation (8) is analytically derived in Nørrelykke & Flyvbjerg (2011).

We train RNNs with infinite number of training samples, which are generated online and divided into batches of 32 sequences. RNNs, including fully connected RNN, GRU and LSTM, are all with 32 hidden units and `tanh` activation. They are trained with momentum optimizer (Qian, 1999) for 20000 steps, with momentum $= 0.9$ and gradient norm being clipped at 5.0. Learning rate for training is exponentially decayed in a stair-case fashion, with initial learning rate $10^{-4}$, decay rate 0.9 and decay steps 2000.

The mutual information estimators, with learned critics, are trained for 200000 steps with Adam optimizer (Kingma & Ba, 2015) at a flat learning rate of $10^{-3}$. The training batch size is 256, and the validation and evaluation batch sizes are 2048. We use early stopping to deal with overfitting. The training is stopped when the estimation on validation set does not improve for 10000 steps, or when it drops by 3.0 from its highest level, whichever comes first. We use separable critics (Poole et al., 2019) for training the estimators. Each of the critics is a three-layer MLP, with $[256, 256, 32]$ hidden units and [ReLU, ReLU, None] activations. The weights for each layer are initialized with Glorot uniform initializer (Glorot & Bengio, 2010), and the biases are with He normal initializer (He et al., 2015). For the minibatch upper and lower bounds, they are estimated on batches of 4096 sequences.

To train the critics, we feed 100-step BHO sequences into trained RNN to get RNN hidden states and conditional distribution parameters. From each sequence, we use last 36 steps for the inputs to the estimators, where first 18 steps as $x_{\text{past}}[t]$, $t = [1, 2, \ldots 18]$, and the other 18 steps as $x_{\text{future}}[t]$, $t = [19, 20, \ldots 36]$. The hidde state $z_{18}$ is extracted at the last time step of $x_{\text{past}}$.

## A.3 TRAINING DETAILS FOR VECTOR DRAWING DATASET

We train decoder-only SketchRNN (Ha & Eck, 2017) on Aaron Koblin Sheep Dataset, as provided in `https://github.com/hardmaru/sketch-rnn-datasets/tree/master/aaron_sheep`. The SketchRNN uses LSTM as its RNN cell, with 512 hidden units.

For RNN training, We adopt the identical hyper-parameters as in `https://github.com/tensorflow/magenta/blob/master/magenta/models/sketch_rnn/model.py`, except that we turn off the recurrent drop-out, since drop-out masks out informations and will interfere with noise injection.

For mutual information estimations, we use the identical hyper-parameters as descibed in Appendix A.2, except that: the evaluation batch size for critic based estimator, InfoNCE, is set to be 4096, and 16384 for minibatch bounds; early stopping criteria are changed to that either the estimation does not improve for 20000 steps or drops by 10.0 from its highest level, whichever comes first.

Due to the limitation of the sequence length of Aaron's Sheep, we use the samples with at least 36 steps long. The $x_{\text{past}}$ and $x_{\text{future}}$ are split at the middle of the sequences, and each with 18 steps.

Due to the limitation of the dataset size of Aaron's Sheep, we augment the dataset with randomly scale the stroke by a factor sampled from $\mathcal{N}(0, 0.15)$ for each sequence to generate a large dataset. Figure 4 (Right) shows that the augmentation helps in training the estimator.

## A.4 EFFECT OF CHANGING TRAINING OBJECTIVES: MAXIMUM LIKELIHOOD AND CONTRASTIVE LOSS TRAINING

To access whether the observed suboptimality in the information plane was due to the maximum likelihood (MLE) objective itself, we additionally trained constrastive predictive coding (CPC) models (Oord et al., 2018). We used the identical model architecture as described in the Section 2.3. The CPC loss lower bounds the mutual information between the current time step and $K$ steps into the future. For our experiments on the Brownian harmonic oscillator (BHO), we look into $K = 30$ steps future, and use a linear readout from the hidden states to a time-independent embedding of the

inputs. As shown in Figure 10, we found that models trained with CPC loss had similar frontiers as those trained with MLE. The loss function does not appear to have a substantive effect.

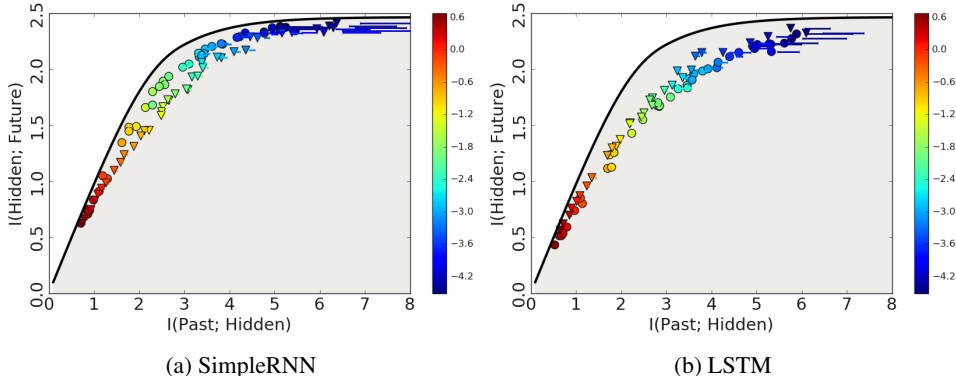

(a) SimpleRNN          (b) LSTM

Figure 10: The impact of training objectives on BHO dataset for SimpleRNN (**Left**) and LSTM (**Right**). Models trained with maximum likelihood estimations are marked with ∘, and models trained with contrastive loss are marked with ▽. The color bar shows the noise level in $\log_{10}$ scale.

## A.5 EFFECT OF DROPOUT

Dropout (Srivastava et al., 2014) is a common method applied on neural network training to prevent overfitting. We trained fully connected RNNs and LSTMs with different levels of dropout probability. As shown in Figure 11, we find that RNNs trained with dropout extract less information than the ones without it, but the information frontier of the models does not change, when we sweep dropout rate and additive noise. In other words, we can find models that are equivalent to dropout models by simply adding Gaussian noise to the output at training time.

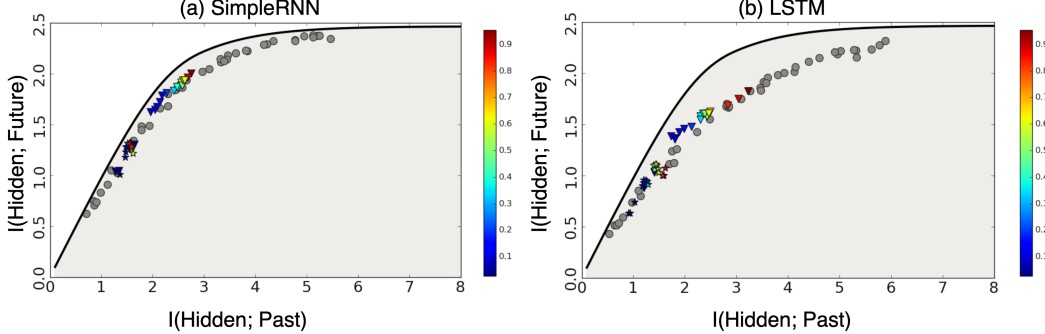

Figure 11: The impact of dropout on predictive information capacity for SimpleRNN (**Left**) and LSTM (**Right**). Grey ∘ marks the result of stochastically trained RNN as described in Figure 2. Colored marks the result for stochastically trained RNN with gaussian noise and different keep rate on RNN outputs, with the keep rate colored in heat. ▽ ones are with noise level 0.1, and ⋆s are with 0.5.

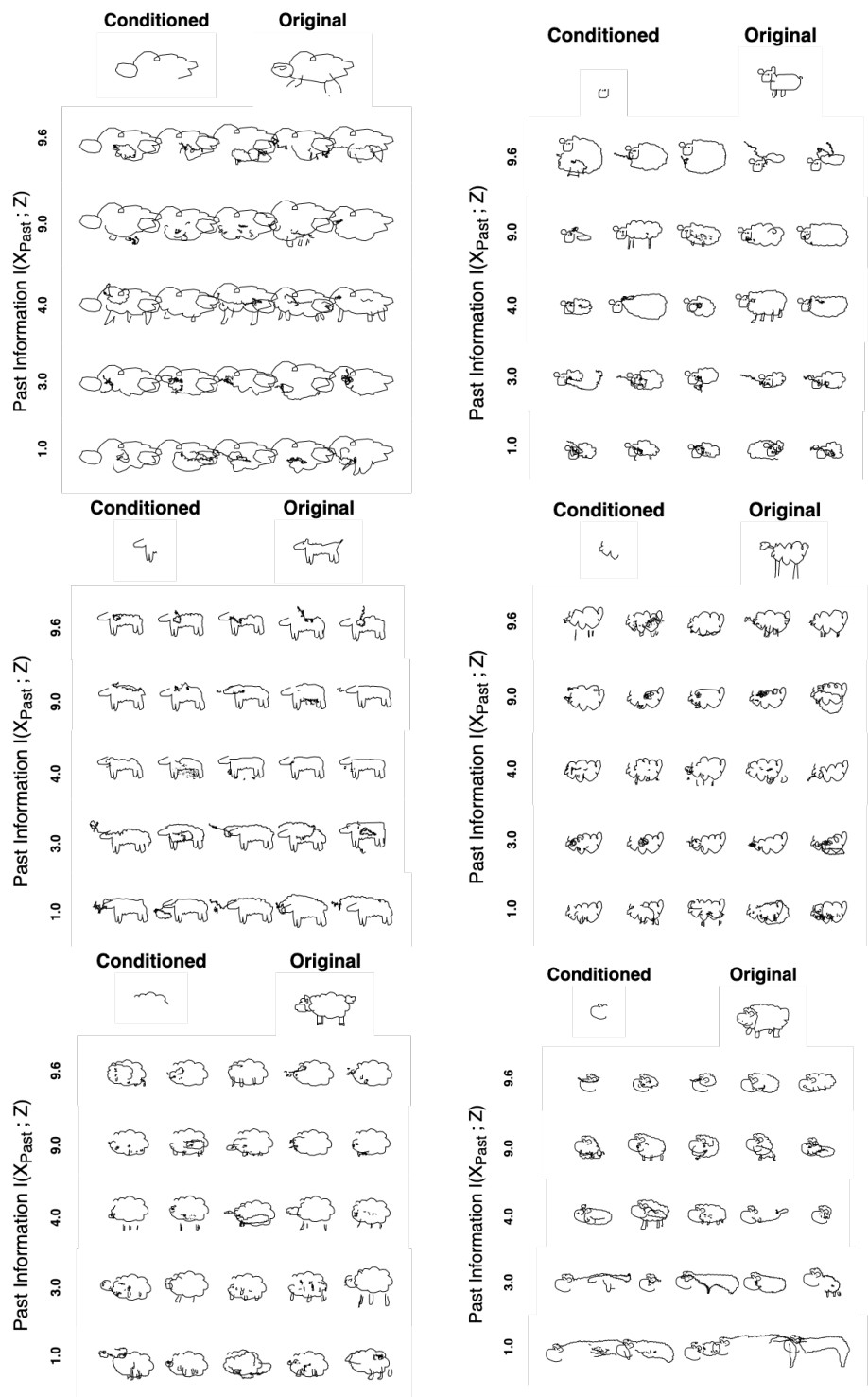

Figure 12: Conditionally generated samples from models with different levels of past information.

