# OpenReview forum: "On Predictive Information Sub-optimality of RNNs"
_ICLR.cc/2020/Conference — Reject_

### Official Review · AnonReviewer1 · 2019-10-15
**Official Blind Review #1**

**Rating:** 3

**Review:**

Summary:

The paper investigate how optimal recurrent neural networks (RNNs) are at storing past information such that it is useful for predicting the future. The authors estimated optimality in terms of mutual information between the past and the future. If the RNN was able to retain MI between the past and the future, it then has kept optimal information from the past for predicting the future. The experiments suggest that RNNs are not optimal in terms of prediction of the future. It also suggest that this is due to the maximum likelihood training objective.


Comments for the paper:

1. Overall, the paper is a very interesting read and it explores and analyzes RNN under a different light. It answers a fundamental question about RNN training.

2. There are a few things that would be nice to clarify. At the end of P3, the authors mentioned that the stochastic RNNs are trained either by a). deterministically during training and noise added during test or b) noise added during training and test. It is not very clear to me how the authors trained stochastic RNNs deterministically during training. It would be nice if this can be clarified.

3. I am also curious how this compares to the training methods for example used in https://papers.nips.cc/paper/7248-z-forcing-training-stochastic-recurrent-networks.pdf. It seems that this would also help with retaining RNN optimality in terms of predicting the future. it would be interesting to include a comparison to this method for example.

Overall an interesting paper. However, I think a few things could be improved and I would be willing to rise the score if the authors could addressed the above points.


**Experience Assessment:**

I have published one or two papers in this area.

**Review Assessment: Checking Correctness Of Derivations And Theory:**

I assessed the sensibility of the derivations and theory.

**Review Assessment: Checking Correctness Of Experiments:**

I assessed the sensibility of the experiments.

**Review Assessment: Thoroughness In Paper Reading:**

I read the paper at least twice and used my best judgement in assessing the paper.

---

> ### Author Response · Authors · 2019-11-15
> **Reply to Review #1**
>
> Thank you for your review and thoughtful feedback!
>
> --It is not very clear to me how the authors trained stochastic RNNs deterministically during training.
>
>      We compared two different setups. In the first setup, we trained deterministic RNNs, and then added noise post-hoc at test-time, i.e. the model used to train and the model used to test are different. In the second setup, we trained stochastic RNNs with noise at training time, and added that same amount of noise at test-time, thus the train and test models are the same. These are two different ways to create a stochastic RNN at test-time. As one would expect, models trained with noise perform better when evaluated with noise, while models trained deterministically perform worse when noise is added. We included the first case primarily as a baseline, and as a means to estimate information in deterministically trained RNNs where it would otherwise be intractable.
>
> --Regarding: comparison to other methods
>
>       We trained RNNs and LSTMs with output dropout (in Appendix A.5) and with the CPC objective instead of MLE (in Appendix A.4), while we were not able to compare to Z-Forcing in the short rebuttal period. We find that RNNs trained with dropout extract less information than RNNs without dropout, but the frontier of models when we sweep dropout rate and additive noise does not change. In other words, we can find models that are equivalent to adding dropout just by increasing the amount of Gaussian noise on the output. We have added an additional figure and discussion to the appendix including models trained with dropout.
>
>       Similarly, we find that models trained with CPC perform similarly to models trained with MLE, and are not more optimal on the information plane. However, this may be due to the toy BHO dataset having Markovian dynamics, meaning that optimizing for one step ahead prediction with MLE is sufficient to maximize mutual information with the future of the sequence. Additional details and a figure can be found in the Appendix A.4 and Figure 10 in the paper.
>
>       We hope that these additional experiments and improvements to the clarity of the text have addressed all of your concerns.

---

### Official Review · AnonReviewer2 · 2019-10-24
**Official Blind Review #2**

**Rating:** 6

**Review:**

This manuscript shows that good ability to compress past information in RNNs help them to predict the future, and that improving upon this ability leads to more useful RNNs. The manuscript first adapts modern mutual-information estimators to mini-batch settings in order to measure the information that an RNN has on the past. It then considers stochastic training, adding Gaussian noise to the hidden states during the training of the RNNs to limit past information. A significant section is dedicated to an empirical study that shows that classically-train MLE RNNs lead to internal representations with a suboptimal mutual-information to the past and the future. For LSTM and GRU architecture, stochastic training actually significantly helps. Experiments on applications such as synthetizing hand-drawn sketches suggest that stochastic training leads to more useful RNNs.

This work has interesting observations and makes a credible case. The stochastic training does seem useful. However, I would like to understand better how it connects to the set of publications discussing dropout in RNNs. It is already known that stochastic perturbations during training help. In addition, the way stochastic training is introduced in this paper make it seem a bit contradictory with the fact that it actually helps generalization. I have the feeling that the benefit that is not understood and that intuitions that arise from the manuscript may not be as useful as we would like.


**Experience Assessment:**

I have read many papers in this area.

**Review Assessment: Checking Correctness Of Derivations And Theory:**

I assessed the sensibility of the derivations and theory.

**Review Assessment: Checking Correctness Of Experiments:**

I assessed the sensibility of the experiments.

**Review Assessment: Thoroughness In Paper Reading:**

I read the paper at least twice and used my best judgement in assessing the paper.

---

> ### Author Response · Authors · 2019-11-15
> **Reply to Review #2**
>
> Thank you for your careful reading and feedback! We agree that stochastic training is a useful component of limiting information in RNNs, and performed additional experiments to address the connection with dropout as shown in Appendix A.5 and Figure 11.
>
> --The way stochastic training is introduced in this paper make it seem a bit contradictory with the fact that it actually helps generalization
>
>       While stochastic training may limit performance on the training set (i.e. reducing I(hidden state; labels)), this often acts to regularize the model in a way that is beneficial for generalization to the test set, as we show in our experiments on the sketch dataset where training data is limited. Previous work on VIB (Alemi et al., 2017) has also shown how stochastic bottlenecks can improve generalization in the classification setting.
>
> --Intuitions that arise from the manuscript may not be as useful as we would like.
>
>       As we emphasize throughout the manuscript, we find that RNNs trained without noise extract too much information about the past, and this can be harmful when training models on small datasets. Our simple and intuitive procedure of adding noise to hidden states to discard information presents one mechanism for limiting the capacity of hidden states, and our analysis in the information plane helps to reveal the tradeoffs between extracting information about the past and being able to predict the future of a sequence.
>
> Reference:
>     Alexander A. Alemi, Ian Fischer, Joshua V. Dillon, Kevin Murphy. Deep Variational Information Bottleneck. ICLR 2017. https://arxiv.org/abs/1612.00410

---

### Official Review · AnonReviewer4 · 2019-11-04
**Official Blind Review #4**

**Rating:** 3

**Review:**

This paper certainly poses an interesting question: How well do RNNs compress the past while retaining relevant information about the future. In order to quantitatively answer this question, the authors suggest to look at (the optimal solutions of) the Information Bottleneck Lagrangian (IBL). The investigated RNNs need to solve the task of next-step prediction, which can be used to evaluate the IBL. In the paper, the (deterministic) hidden state h is transformed through simple additive Gaussian noise into a stochastic representation which then is utilized to compute the IBL. In general the IBL is not tractable and hence the paper uses approximate computations.

I definitely like the underlying question of the paper. Yet, to me it seems not ready for publication. For one, the presented experimental results look interesting but the suggested method for improvement through adding noise to the latents (during training) is too much of handwaving for such a fundamental problem. Second, shouldn't the results on the BHC be quite surprising in terms of LSTMs performance? Why is that, usually LSTM (or GRU) deliver excellent performance in typical (supervised) sequential tasks. Third, the task itself seems not well described, it seems to be next-step prediction, but how are more future predictions generated -- these seem to be not considered in the equation, but probably should when talking about 'retaining relevant information for the future'? Fourth, recently some researchers started to question whether 'reconstruction' is a good idea in order to learn generative-like models, for example you cite van den Oord 2018. How would such models perform in your metric.

A final remark with respect to your citation for eq. 4, I think you meant Barber, Agakov, "The IM algorithm...", 2003?

**Experience Assessment:**

I have read many papers in this area.

**Review Assessment: Checking Correctness Of Derivations And Theory:**

I assessed the sensibility of the derivations and theory.

**Review Assessment: Checking Correctness Of Experiments:**

I assessed the sensibility of the experiments.

**Review Assessment: Thoroughness In Paper Reading:**

I read the paper at least twice and used my best judgement in assessing the paper.

---

> ### Author Response · Authors · 2019-11-15
> **Reply to Review #4**
>
> Thank you for your careful reading and thoughtful feedback!
>
> --“ the suggested method for improvement through adding noise to the latents (during training) is too much of handwaving for such a fundamental problem"
>
>       Adding noise to a bounded representation is a tractable and effective method for introducing a bottleneck into a hidden state with a bounded activation. Furthermore, for any level of additive noise, we can think of this process as introducing a hard constraint on the amount of information that can be stored in the hidden state, thus we’re solving a constrained optimization problem instead of the typical Lagrangian formulation in many applications of Information Bottleneck. While there are many other approaches for introducing bottlenecks (e.g. Dropout, computing a variational upper bound on the information and pushing that down), we found that adding noise was a simple strategy that was effective and interpretable.
>
> --Shouldn't the results on the BHC be quite surprising in terms of LSTMs performance?
>
>       RNNs and LSTMs trained with deterministic hidden states perform well at next step prediction, but they do this by extracting far more information than is needed to solve the task. For large datasets this may not be an issue, but as we show in the QuickDraw experiments, when the data is limited, constraining the amount of information in the hidden state can be useful for improving generalization.
>
> --How are more future predictions generated.
>
>       To evaluate mutual information between the hidden state and future states of the world, we do not need to generate future samples from our model, we only need hidden state samples paired with the true future observations. Furthermore, for a Markov process we only need to look a single step into the future to evaluate information with the infinite future.
>
> --Recently some researchers started to question whether 'reconstruction' is a good idea in order to learn generative-like models, for example you cite van den Oord 2018. How would such models perform in your metric.
>
>       Thank you for this suggestion! We have performed additional experiments training RNNs and LSTMs with CPC. The model architecture is identical to our initial experiments, but instead of training with MLE we train with the CPC loss, which estimates the mutual information between the current timestep and K steps into the future. For our experiments on the BHO, we looked up to 30 steps into the future, and used a linear readout from the hidden state to a time-independent embedding of the inputs. We found that models trained with CPC had a similar frontier as those trained with MLE, and added a figure and additional details to Appendix A.4 and Figure 10 in the updated paper.

---

### Decision · Program_Chairs · 2019-12-19

**Decision:**

Reject

**Comment:**

Nice start but unfortunately not ripe.  The issues remarked by the reviewers were only partly addressed, and an improved version of the paper should be submitted at a future venue.